# Chemical Composition of *Tagetes patula* Flowers Essential Oil and Hepato-Therapeutic Effect against Carbon Tetrachloride-Induced Toxicity (In-Vivo)

**DOI:** 10.3390/molecules27217242

**Published:** 2022-10-25

**Authors:** Hanan Y. Aati, Mahmoud Emam, Jawaher Al-Qahtani, Sultan Aati, Abdulrahman Aati, Juergen Wanner, Mohamed M. Seif

**Affiliations:** 1Department of Pharmacognosy, College of Pharmacy, King Saud University, P.O. Box 22452, Riyadh 11495, Saudi Arabia; 2Phytochemistry and Plant Systematics Department, National Research Centre, Giza 12622, Egypt; 3UWA Dental School, University of Western Australia, 17 Monash Avenue, Nedlands, WA 6009, Australia; 4Rokn Al-Madaein Pharmaceutical Warehouse Co., P.O. Box 2990, Riyadh 11495, Saudi Arabia; 5Kurt Kitzing Co., Hinterm Alten Schloss 21, D-86757 Wallerstein, Germany; 6Toxicology and Food Contaminants Department, Food Industries and Nutrition Research Institute, National Research Centre, Giza 12622, Egypt

**Keywords:** CCL_4_, essential oil, GC/MS, in vivo, hepato-therapeutic, HD, *Tagetes patula*

## Abstract

The liver is a crucial organ among body organs due to its wide functions, in particular, detoxification and metabolism. Exposure to detrimental chemicals or viral infections may provoke liver dysfunction and ultimately induce liver tissue damage. Finding natural substances for liver disease treatment to overcome the conventional treatments’ side effects has attracted the attention of researchers worldwide. Our current work was conducted to investigate the hepato-therapeutic activities of essential oil (EO) isolated from *Tagetes patula* flowers. EO was extracted using the hydro-distillation (HD) technique and its chemical composition was identified by GC/MS. Then, the hepatic treatment potential of extracted EO was evaluated in vivo against CCL_4_ in rats. HD of *T. patula* flowers yielded highly chemical constituents of EO along with significant antioxidant potential. A coherent molecular network was fashioned via the Global Natural Products Social Molecular Networking (GNPS) to visualize the essential components and revealed that the sesquiterpene (*E*)-*β*-caryophyllene was the most predominant volatile constituent which accounted for 24.1%. The treatment of CCL_4_ led to significant induced oxidative stress markers malonaldehyde, total protein, and non-protein sulfhydryl, as well as elevated serum aminotransferase, gamma-glutamyl transferase, alkaline phosphatase, and bilirubin. In addition, it disrupted the level of lipid profile. The post-treatment using *T. patula* EO succeeded in relieving all toxic effects of CCl_4_ and recuperating the histopathological signs induced by CCL_4_. Silymarin was used as a standard hepatoprotective agent. The obtained results demonstrated that the extracted EO exerted high protective activities against the toxicity of CCL_4_. Moreover, the *T. patula* flowers EO can be used as a natural remedy to relieve many contemporary liver diseases related to oxidative stress.

## 1. Introduction

The liver is a main organ in the body that plays a vital role in the metabolism, detoxification, and removal of different toxic chemicals from the body. It regulates different physiochemical functions, such as oxidation, reduction, hydroxylation, hydrolysis, conjugation, sulfation, and acetylation [1]. Currently, liver diseases pose serious health issues worldwide. Environmental pollutants, food contaminants, and some chemical drugs are associated with liver damage, via producing free radicals agents when metabolized in the liver [2,3,4]. Carbon tetrachloride (CCl_4_) is a compound that is most commonly used to induce liver injuries in experimental animals model [5,6]. It is a well-known hepatotoxic agent, which may provoke liver injury through multiple mechanisms, including oxidative stress, inflammatory response, and apoptosis [7,8]. In hepatocytes, CCL_4_ is converted to trichloromethyl free radicals that can inaugurate cell damage by reacting with cell macromolecules [9] causing oxidative stress, inflammation, and cellular necrosis, which leads to hepatocellular damages, such as fibrosis, cirrhosis, and atrophy [10].

Due to the potential adverse effects of chemical hepatic drugs, contemporary research focuses on finding alternative natural hepatoprotective and hepato-therapeutic products from natural resources, such as plants, with high efficacy for the treatment and protection from liver disorders [11,12]. Essential oil (EO) extracted from different parts of plants or their exudates have major therapeutic effects in aromatherapy [13].

*Tagetes patula* (French marigold), which belongs to the family composite, is vastly recognized for its phytochemical and therapeutic properties. Traditionally, the plant is used to treat many symptoms and diseases, such as cough, colic, constipation, diarrhea, rheumatism, and eye problems. *T. patula* is known for its antimicrobial, antioxidants, antiseptic, blood purifying, and diuretic activities. In addition, its flowers are edible and used in refreshing drinks [10]. Different parts of the plant contain many phytochemical components, such as carotenes, trepans, steroids, flavonoids, and thiophenes [14].

Few studies were performed on the chemical composition and biological action of *T. patula* essential oil (EO). Therefore, this current study particularly aimed at the extraction of EO from the flowers of the *T. patula* that exist in the Kingdom of Saudi Arabia. Then, an evaluation of its phytochemical constituent content is carried out as well as an appraisal of its efficacy against hepatotoxicity provoked by CCl_4_. To the best of our knowledge, our work is the first time to investigate the hepato-therapeutic properties of the essential oil obtained from the flowers of *T. patula* growing in the Kingdom of Saudi Arabia.

## 2. Materials and Methods

### 2.1. Plant Material, Essential Oil Extraction, and GC/MS Analysis

The fresh flowers (100 g) of *T. patula* were collected during the flowering stage from the FAYFA Mountains, Jizan province, Kingdom of Saudi Arabia, in April 2021. The plant samples were identified by taxonomist Dr. Rajakrishnan Rajagopal, KSU, Riyadh, Saudi Arabia, and the specimen was deposited in the Herbarium, KSU (Voucher #KSU24550). The flower of *T. patula* was extracted by hydro-distillation (HD) as described in [15]. Moreover, the GC/MS analysis and identification were performed using the methods proposed in [15].

### 2.2. Molecular Networking GC Workflow

A molecular network was created using the Library Search/Molecular Networking GC workflow at GNPS [16]. The precursor ion mass tolerance was set to 20,000 Da and the MS/MS fragment ion tolerance to 0.5 Da. Then, a molecular network was created, where edges were filtered to have a cosine score above 0.7 and more than six-matched peaks. Furthermore, edges between two nodes remained in the network only if each of the nodes appeared in each other’s respective top 10 most similar nodes. Finally, the maximum size of a molecular family was set to 100, and the lowest-scoring edges were removed from molecular families until the molecular family size was below this threshold. All of the matches that remained between network spectra and library spectra were required to have a score above 0.85 and at least six-matched peaks. The molecular networks were visualized using Cytoscape software (3.9.0, The Cytoscape Consortium, San Diego, CA, USA) [17].

### 2.3. Hepatoprotection Study of Essential Oil Extracted from T. Patula

#### 2.3.1. Antioxidant Activity

The in vitro antioxidant activity of EO extracted from *T. patula* flower was evaluated based on a 2,2-diphenyl-1-picrylhydrazyl (DPPH) scavenging assay, nitric oxide (NO) radical scavenging assay, and ferric reducing antioxidant power assay (FRAP), according to Koksal et al. [18], Rao [19], and Benzie and Strain [20], respectively.

#### 2.3.2. Biological Evaluation of the Hepatotherapy for EO of *T. Patula* (In Vivo)

##### Acute Toxicity of Extracted EO of *T. Patula*

Acute toxicity of extracted EO of *T. patula* was ascertained orally, in which (OECD, 401) thirty adult Wistar albino rats were treated separately with six nominal experimental concentrations of extracted EO (10, 20, 50, 100, and 200 mg/kg BW). The tests were performed semi-statically for 96 h. Mortality was recorded after 96 h of exposure period. The symptoms of tremors, convulsions, salivation, micturition, defecation, lethargy, sleep and coma, respiration, sedation piloerection, and writhing were recorded during the first 4 h. The recorded mortality data were used to calculate the 96-h lethal dose (LD_50_) value according to Mamza et al. [21].

##### Animals and Experimental Protocol

A total of 40 weaned Wistar albino rats weighing between 180 and 200 g were provided by the Experimental Animal Care Center of the College of Pharmacy, King Saud University, Riyadh. The animals were maintained on a standard chow diet and housed in polycarbonate cages in a room free from any source of chemical contamination, artificially illuminated (12 h dark/light cycle), and thermally controlled (25 ± 2 °C) at the animal facility. All of the animals received humane care in compliance with the guidelines of the Ethics Committee of the Experimental Animal Care Society, College of Pharmacy, King Saud University, Riyadh, Saudi Arabia. Moreover, all of the rats were given 1 week of acclimatization, prior to being randomly allocated to form five groups (eight animals in each group) and treated as follows (illustration in Figure 1).

○**Control group**: Untreated group.○**CCl_4_ group** (CCL_4_): Rats were treated with intraperitoneal CCL_4_ at a dose of 1.25 mL kg^−1^ BW for 15 days.○**CCL_4_/Silymarin group (CCL_4_/SL):** Rats received CCL_4_ at the abovementioned dose for 15 days, and then treated orally with silymarin at a dose of 10 mg kg^−1^ BW for the next 15 days.○**CCL_4_/Essential oil: (CCL_4_/EO-5)**: Rats were exposed to CCL_4_ at a dose of 1.25 mg kg^−1^ BW for 15 days, and then the extracted EO was administered orally by gavage at a dose of 5 mg kg^−1^ BW for the next 15 days.○**CCL_4_/Essential oil: (CCL_4_/EO-10):** Rats received CCL_4_ at a dose of 1.25 mg kg^−1^ BW for 15 days, and then the total phenolic content (TP) of EO was administered orally by gavage at a dose of 10 mg kg^−1^ BW for the next 15 days.

**Figure 1 molecules-27-07242-f001:**
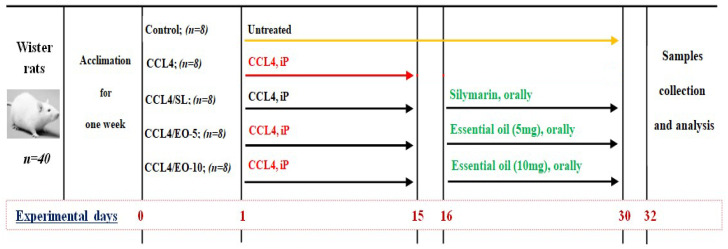
Scheme of the experimental protocol CCL_4_: Carbon tetrachloride; SL: Silymarin; iP: Intraperitoneal.

The dose of EO was selected on the basis of the acute toxicity test, while the intoxicant dose of CCL_4_ was selected according to the previously published studies [22]. The blood specimens were collected from the Orbital sinus of the experimental rats under short anesthesia using isoflurane on the 16th day for Group 2. Moreover, on the 32nd day for groups 1, 3, 4, and 5, serum samples were obtained by centrifugation of the blood samples at 3000 rpm for 15 min. Then, the animals were sacrificed using ether anesthesia and the liver tissue was dissected and used for biochemical and histological inspection.

##### Histopathological Inspection

Tissues excised from formalin-fixed liver were used for paraffin block preparation, then sliced into 4-μm sections. The sections were mounted onto slides, blotted by H&E, and examined using a microscope (Olympus BX51, Tokyo, Japan) with an Olympus E-330 camera [23].

##### Oxidative Stress Biomarkers

Malondialdehyde (MDA) was assayed spectrophotometrically by the reaction with thiobarbituric acid as an indicator of lipid peroxidation according to the procedure reported by Ohkawa et al. [24]. Then, the content of MDA (nmol/g) was calculated by reference to a standard curve of MDA solution. The hepatic non-protein sulfhydryl (NP-SH) was measured according to the methods reported by [25] in tissue homogenates. The absorbance was carried out within 5 min of adding the DTNB solution at 412 nm against a reagent blank. Meanwhile, the total protein (TP) of hepatic was estimated using the method of Lowery [26].

##### Serum Liver Functions Marker

Aminotransferase enzymes GOT and GPT were assessed as stated by Reitman and Frankel [27]. Alkaline phosphatase (ALP), gamma-glutamyl transferase (GGT), and bilirubin were determined according to the report by Otto et al. [28], Whitfield [29], and Doumas et al. [30], respectively. All of the kits were purchased from Roche Diagnostics GmbH, Mannheim, Germany.

##### Estimation of Lipid Profiles

Serum total cholesterol, triglycerides, and high-density lipoproteins (HDL) were estimated in serum samples according to kit instructions. The low-density lipoprotein (LDL) was computed using a standard formula suggested by Davidson and Rosenson [31]:**LDL** = Total cholesterol − [HDL + triglycerides/5] × 100(1)

### 2.4. Statistical Analysis

The data obtained in this study were expressed as mean ± SD. For assessments of the results, one-way analysis of variance (ANOVA) was used, followed by a post hoc Tukey test. The value of *p* < 0.05 was considered as statistically significant between the empirical groups. All the statistical analyses were performed using SPSS software (v.11.5, IBM, Armonk, NY, USA) and the figures were created using the GraphPad Prism 8 (GraphPad Software, San Diego, CA, USA).

## 3. Results

### 3.1. T. Patula EO Chemical Constitutions

The HD technique was used to extract the essential oil of *T. patula* flowers prior to injection into the GC/MS to detect the different volatile skeletons, as shown in Figure 2. The nominated metabolites of the *T. patula* flowers were provided in Table 1. The HD of the *T. patula* flowers produced a yellow oil with a yield of 0.84% *v*/*w*. According to GC/MS results, *T. patula* EO was found to contain 79 constituents, with 89.8% identified constituents (Table 1).

Although (*E*)-*β*-caryophyllene represents the most predominant volatile sesquiterpene constituent (24.1%), the monoterpenoids exhibited the major existing structures (31 structures, 39.24%), as shown in Figure 3. Moreover, phenylpropanoids increased to the third level with 10.13%. On the other hand, the volatile ketones, fatty acid alkyl ester, volatile alcohols, benzene derivatives, and the mixed structures of monoterpenes esters or ketones represented 7.59%, 3.80%, 3.80%, 2.53%, and 2.53%, respectively. Furthermore, the volatiles diterpene, ester, and ether showed the lowest structures with 1.27% for each one of them. Finally, the only acyclic diterpene identified was neophytadiene with 0.66% (Table 1).

As shown in Table 1, the identified volatile compounds can be arranged in descending order as follows: (*E*)-*β*-caryophyllene (24.1%) > 2-undecanone + bornyl acetate (12.2%) > 2-nonanone (9.7%) > camphor (4.4%) > limonene (4.3%) > biphenyl (I.S.) (3.2%) > 2-nonyl acetate + cis-ocimenone (2.3%) > piperitenone (2.0%) > cis-davanone (1.8%) > germacrene D (1.3%) > δ-cadinene (1.3%) > linalool (1.2%) > carvone (1.2%) > (*Z*)-ethyl cinnamate (1.2%), respectively, depending on their percent areas.

Next, the molecular network (MN) was built from the GC/MS data using the GNPS online plate form to visualize the volatiles and analyze the results. The created MN (Figure 3) demonstrated that the (*E*)-*β*-caryophyllene is the most abundant metabolite.

### 3.2. Antioxidants Activity of EO Extracted from T. Patula

Data summarized in Table 2 revealed that the *T. patula* EO showed high antioxidant activities against DPPH, NO, and FRAP with EC_50_ 29.85 ± 4.53, 33.19 ± 3.8, and 30.22 ± 2.12 µg/mL, respectively compared with the activities of ascorbic acid.

### 3.3. Acute Oral Toxicity of Essential Oil of T. Patula

The effects of extracted TP EO on groups, doses of 100 and 200 mg/kg BW demonstrated symptoms of micturition, defecation sedation, and minor agitation. In Table 3, the mortality was recorded and the values were used to estimate the lethal dose (LD_50_) using the Karbar method. The calculated LD_50_ was estimated as 126 mg/kg BW.

### 3.4. Histopathology Findings

The curative effects of EO extracted from *T. patula* against CCL_4_ toxicities in rats were asserted by the histopathological inspection. The normal architecture of the hepatocytes and clear sinusoids were observed in the hepatic section of the untreated group (Figure 4A). On the other hand, the liver section of CCL_4_-treated rats manifests multiple histological alterations, including altered hepatocyte morphology, loss of cell membrane, increased nuclear size, and connective tissue infiltration with prominent necrosis and vacuoles (Figure 4B). Furthermore, the liver of rats treated with vehicle-treated control exhibits normal architecture hepatocytes, intact cell membrane, and central vein (Figure 4C). Meanwhile, the rats treated with EO after CCL_4_ at 5 and 10 mg/kg BW, showed notable recovery of the liver histopathological signs when compared with the CCL_4_ group (Figure 5E,F). The rats treated with silymarin after CCL_4_ did not manifest any pathological alterations compared with the control slide.

### 3.5. EO Effect on MDA, TP, and NP-SH in CCL_4_-Treated Rats

The malondialdehyde (MDA), non-protein sulfhydryl (NP-SH), and total protein (TP) were assessed in liver tissue as characteristics of oxidative stress. Figure 5 shows that CCL_4_-administration provoked a notable reduction (*p* < 0.05) in TP and NP-SH in the CCL_4_-treated group in comparison to the control group. In contrast, CCL_4_ administration increases lipid peroxidation events, measured as MDA products. Meanwhile, rats treated with EO after CCL_4_ showed significant enhancement in TP and NP-SH levels at the two tested doses. Moreover, significant recovery in lipid peroxidation was similarly noted in the silymarin effect.

### 3.6. EO of T. Patula Effect on Serum Liver Functions in CCL_4_-Treated Rats

CCL_4_ induced pathologic changes in the liver, including hepatomegaly and serologic changes, along with increased activities of GOT, GPT, bilirubin, and GGT. Figure 6 shows that CCL_4_ treatment significantly (*p* < 0.05) increased the activities of GOT, GPT, GGT, and bilirubin, while the post-administration of EO (5 and 10 mg/kg BW) after CCL_4_ significantly (*p* < 0.05) mitigated these functional markers toward near-normal levels. *T. patula* EO at a dose of 10 mg/kg BW was more effective than the lowest dose of 5 mg/kg BW. A similar biological effect was noticed in rats treated with silymarin.

### 3.7. EO Effect on Lipid Profile in CCL_4_-Treated Rats

To evaluate the palliation impacts of *T. patula* EO on lipid profile markers in CCL_4_-treated rats, the serum cholesterol, TriG, LDL, and HDL were estimated. The deduced results in Figure 7 illustrated that the CCL_4_ treatment resulted in a marked increase in total cholesterol, TriG, and LDL with a significant decrease in HDL levels compared with the untreated group. In contrast, post-treatment of extracted EO after CCL_4_ showed significant (*p* < 0.05) restoration in tested lipid profile markers when compared with the control group, indicating improvement of the lipid metabolism in EO post-treated rats at two doses. The TriG levels were significantly lower in rats treated with 10 mg of EO than those treated with silymarin following CCL_4_ treatment.

## 4. Discussion

The liver recreates crucial functions in the metabolism and biotransformation of xenobiotics. On account of its position amidst the intestinal tract and circulatory system, it receives extensive quantities of xenobiotics and nutrients absorbed via the digestive tract and the portal vein, becoming the target organ of sundry categories of natural or synthetic toxicants [1].

This study aimed to extract the essential oil from the flower of *T. patula* and assess its protective utility against CCl_4_-induced liver damage in rats. *T. patula* EO was extracted using the HD, common, reliable, sensitive, and accurate technique [32]. The phytochemical composition results showed that the *T. patula* EO contains 79 compounds that represent 89.8% total volume of the extracted EO. The primary groups of detected constitutes are monoterpenes and sesquiterpenoids, which represented 39.2% and 25.32%, respectively. These were documented to exert many biological properties, including antioxidant and anti-inflammatory properties [33]. Furthermore, the prime compound in the extracted EO is *E-β*-caryophyllene, which represents 24.1% of the total extracted EO. This component is documented to have good antioxidant and anti-inflammatory activities [34].

According to the present findings, CCl_4_ caused liver disorders in rats, which were portrayed by oxidative stress, pathological tissue damage, elevated liver enzymes, and disruption of lipid metabolism. These results are in agreement with the results of [35,36,37,38]. The toxic mechanism of CCl_4_ occurs during liver metabolism, wherein the cytochrome P450 (CYP) enzyme converts CCl_4_ to the trichloromethyl radical (CCl_3_) [39]. This process impairs many vital cellular processes, induces vast cell damage, and causes the release of the aminotransferase enzyme into blood circulation. In hepatic apoptosis, the synthesis of cellular phospholipids refers to the amalgamation of phospholipids into lipoproteins, conducting the assembly of triglycerides [40].

Moreover, CCl_4_ metabolism in the liver caused oxidative stress and suppressed the antioxidant defense system [41,42]. The current results showed that the CCl_4_ treatment resulted in significant alleviation of MDA, NP-SH, and TP. Lipid peroxidation is the principal mechanism of hepatic injury [43]. In the liver, CCl_4_ transforms to trichloromethyl (CCl_3_) under the catalytic activity of cytochrome P450. CCl_3_ is a free radical, that mainly can react with oxygen to produce toxic trichloromethyl peroxyl (CCl_3_O_2_) radicals [39]. The outcome CCl_3_O_2_ has the prospect to bind to myriad proteins or lipids and induce the lipids’ peroxidation [44,45]. In the present investigation, treatment with CCL_4_ led to the initiation of the degradation of lipids in the cellular membrane. This caused the generation of MDA products that result in a loss of cell membrane integrity and liver injury [46]. The treatment by the extracted EO from *T. patula* markedly lessened the production of the (MDA) in CCl_4_-treated rats. This shows that the administration of *T. patula* essential oils at two tested doses efficaciously minimized lipid peroxidation which is influenced by CCl_4_. These results are harmonious with results published by Riaz et al. [47] as well as Singh and Thakur [48].

Furthermore, the decreased NP-SH content is ideal evidence of hepatotoxicity. The current results showed a marked reduction of the liver NP-SH content in CCL_4_ rats, which could induce further damage and dysfunction of the liver. The treatment by EO of *T. patula* or silymarin significantly re-increased the level of NP-SH in liver tissues indicating a therapeutic potential of the extracted EO.

The shortage of TP is also an indicator of hepatotoxicity. This decrease in total protein could trigger hydration, which is hurtful to cellular homeostasis. This will negatively impact the metabolic activities of the liver, and thus body health [49]. Enhancement of the levels of TP by EO at two tested doses and silymarin denotes a lessening of the oxidative stress, and thus mitigation of hepatic toxicity. In addition to the mitigation of liver toxicity, the extracted EO showed good in vitro antioxidants assays (DPPH, FRAP, and NO). The antioxidant activity of the *T. Patula* EO may be attributed to its constituents and their antioxidant potency.

As stated in the current findings, a significant increase in the liver damage markers was noted in the CCL_4_-treated rats. The excess serum GOT, ALP, GGT, GPT, and bilirubin levels were due to hepatocytes damage [50]. Particularly, the excessive release of bilirubin into the serum of CCL_4_-treated rats has demonstrated the decreased ability of the liver for bile extraction [51].

Therefore, the elevation of the levels of these biochemical markers is a straightforward repercussion of alterations in the hepatic tissue’s structural integrity, which also has been confirmed by histological findings. The post-administration of the extracted oil at two tested doses significantly succeeded in protecting the liver and diminished the hepatic damage markers. These results are congruous with the previously published results regarding the protection of the liver from oxidative stress caused by CCl_4_ using different agents, such as gallic acid and docosahexaenoic acid [22,52,53]. The lipid profile markers are the reliable biomarkers for investigating liver health. The activity of serum lipid profiles, such as triglycerides, total cholesterol, HDL, and LDL was significantly elevated in CCL_4_-treated rats, indicating deterioration in hepatic functions due to the damage caused by CCl_4_ metabolites.

According to our results, liver protection against CCl_4_ has been achieved by administration of EO of *T. patula* at two tested doses through stimulating the regeneration of liver cells or via the enhancement of the antioxidants system, thus scavenging the formed free radicals and preventing their reaction. The hepatoprotective action of the essential oil isolated from *T. patula* can be attributed to its content of monoterpenes and sesquiterpene compounds. Monoterpenes and sesquiterpenes were reported to have manifold pharmacological influences, such as antioxidant and anti-inflammatory activities [54]. According to the chemical composition results, (*E*)-*β* caryophyllene (24.1%) is the major compound in the *T. patula* EO. This sesquiterpene component was documented as a hepatoprotective component against CCL_4_ via exerting its antioxidant and anti-inflammatory effects [34,55]. The hepato-therapeutic effects of *T. patula* EO were associated with mitigation of the oxidative stress in CCL_4_ rats treated by *T. patula* EO compared with the rats treated with CCl_4_ only.

## 5. Conclusions

CCl_4_ is a well-known liver toxic, and commonly used in hepatotoxic models. With increasing cases of liver diseases, the identification, evaluation, and preparation of hepatoprotective drugs from plant sources has become an impressive approach. The extracted essential oil from the flower of *T. patula* yielded 0.43% *w*/*v*. Seventy-nine components were identified by GC/MS analysis and these components represent 89.8% of the oil components. Monoterpenes were the major components of the oil, representing 39.24% followed by sesquiterpenes at 25.32%. The *T. patula* EO showed high antioxidant activities toward DPPH scavenging assay, NO, and FRAP. The sesquiterpene of (*E*)-*β-*caryophyllene as the most predominant volatile constituent among the flower accounted for 24.1%. The impact of the EO was noted as reducing the MDA level toward normal levels. Moreover, restoring TP and NP-SH groups was superior to the effect parallel with the silymarin treatment group. On other hand, the histopathological study showed complete recovery of hepatic tissues in the group treated with CCl_4_ and 10 mg/kg BW *T. patula* EO. Additionally, *T. patula* EO administration restores liver functions and maintains lipid profile at two tested doses. These results suggest that *T. patula* EO can be used to protect and enhance the recovery of the liver to overcome the adverse side effects of some drugs as well as food toxic contaminants.

## Figures and Tables

**Figure 2 molecules-27-07242-f002:**
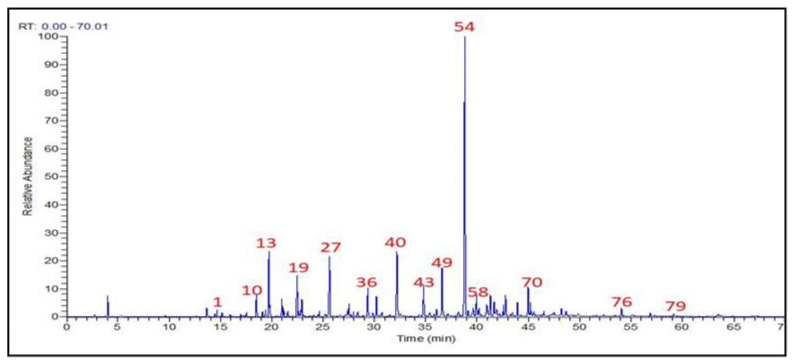
Total ion chromatogram (TIC) for GC/MS of *T. patula* flowers volatile constituents.

**Figure 3 molecules-27-07242-f003:**
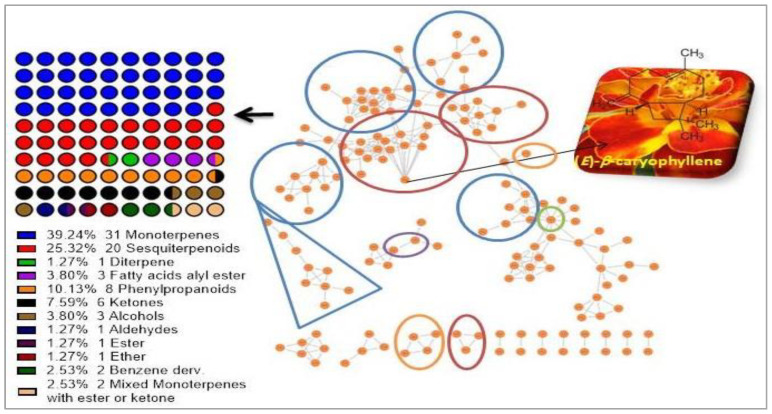
Molecular network of the GC/MS analysis of the identified volatile constituents from *T. patula*, and the percent of metabolites classes. All nodes are labeled with the retention time from the GNPS-GC/MS spectral libraries.

**Figure 4 molecules-27-07242-f004:**
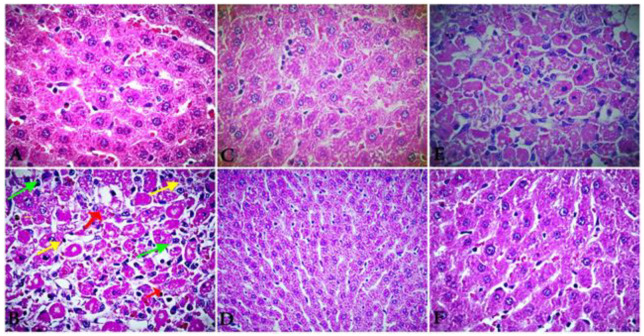
A representative photomicrograph of a section of the liver tissues of rats exposed to CCL_4_ (1.25 mg/kg BW) for 15 days, followed by silymarin (10 mg/kg BW) and extracted EO at two doses (5 and 10 mg/kg BW) for 15 days (E&F). Control rats ((**A**), 100 μm) show a normal architecture of hepatocytes and intact cell membrane. ((**B**), 100 μm) section of liver of rats treated by CCL_4_ shows altered hepatocyte morphology, with loss of cell membrane, increased nuclear size, and connective tissue infiltration with prominent necrosis and vacuoles. ((**C**), 100 μm) section of liver of rat’s vehicle-treated control exhibits normal architecture hepatocytes, intact cell membrane, and central vein. ((**D**), 100 μm) section of liver of rats treated by CCL_4_ (1.25 mg/kg BW) for 15 days and followed by silymarin for the next 15 days exhibits potent hepatoprotective activity with radially arranged hepatocytes similar to the control with intact cell membrane, as well as infiltration of cells formation of normal hepatic cards and absence of necrosis and vacuoles. ((**E**), 100 μm) section of liver of rats treated by CCL_4_ (1.25 mg/kg BW) and followed by extracted EO (5 mg/kg BW) exhibits moderate hepato-therapeutic activity with radially arranged hepatocytes similar to the control with intact cell membrane and infiltration of cells. ((**F**), 100 μm) section of liver of rats treated by CCL_4_ (1.25 mg/kg BW) and followed by extracted EO (10 mg/kg BW) exhibits highly hepato-therapeutic effect with radially arranged hepatocytes similar to the control and silymarin with intact cell membrane and infiltration of cells formation of normal hepatic cards, as well as absence of necrosis and vacuoles.

**Figure 5 molecules-27-07242-f005:**
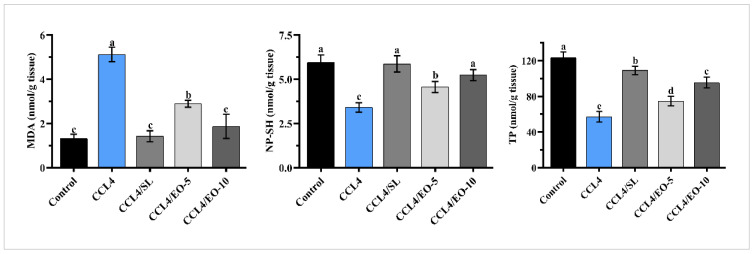
Effect of EO extracted from *T. patula* on MDA, TP, and NP-SH in liver of rats treated with CCL_4_. Data represented as mean ± SD (*n* = 6). CCL_4_: Carbon tetrachloride; SL: Silymarin; EO-5: Essential oil (5 mg/kg BW); EO-10: Essential oil (10 mg/kg BW); MDA: Malondialdehyde; TP: Total protein; NP-SH: Non-protein sulfhydryls. Different letters indicate statistically significant differences between groups at (*p* < 0.05).

**Figure 6 molecules-27-07242-f006:**
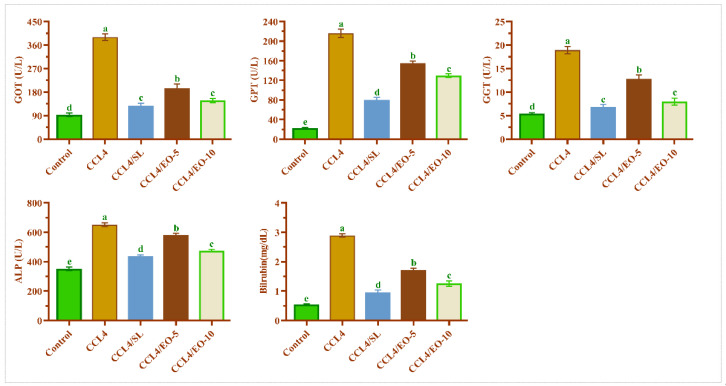
Effect of EO extracted from *T. patula* on liver function markers of rats treated with CCL_4_. Data represented as mean ± SD (*n* = 6). CCL_4_: Carbon tetrachloride; SL: Silymarin; EO-5: Essential oil (5 mg/kg BW); EO-10: Essential oil (10 mg/kg BW); GOT: Glutamic oxaloacetic transaminase; GPT: Glutamic pyruvic transaminase; GGT: Gamma-glutamyl transpeptidase; ALP: Alkaline phosphatase. Different letters indicate statistically significant differences between groups at (*p* < 0.05).

**Figure 7 molecules-27-07242-f007:**
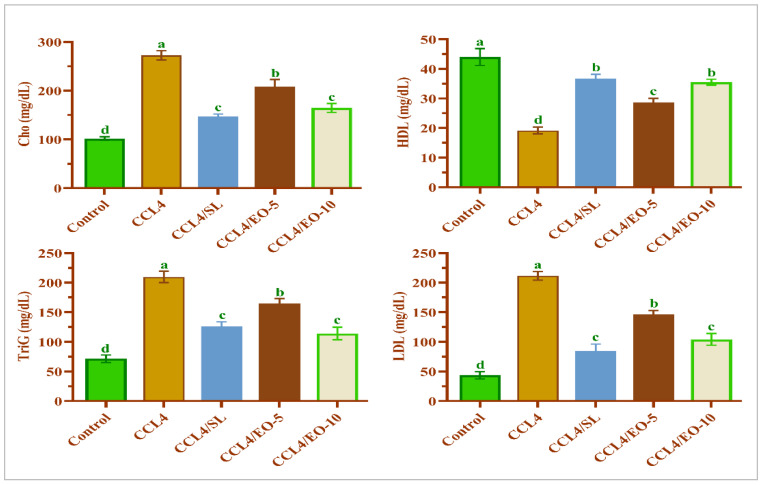
Effect of EO extracted from *T. patula* on lipid profile of rats treated with CCL_4_. Data represented as mean ± SD (*n* = 6). CCL_4_: Carbon tetrachloride; SL: Silymarin; EO-5: Essential oil (5 mg/kg BW); EO-10: Essential oil (10 mg/kg BW); Cho: Cholesterol; TriG: Triglycerides; HDL: High-density lipoprotein; LDL: Low-density lipoprotein. Different letters indicate statistically significant differences between groups at (*p* < 0.05).

**Table 1 molecules-27-07242-t001:** Phytochemical compositions of *T. patula* flowers volatile constituents.

No.	Apex RT	RRT	Compound	RI	% Area	NP Class
1	14.73	0.38	*α*-thujene	932	0.3	Monoterpene
2	15.08	0.39	5-methyl-3-heptanone	939	tr.	ketone
3	15.17	0.39	*α*-pinene	941	0.2	Monoterpene
4	15.95	0.41	camphene	957	0.1	Monoterpene
5	16.17	0.42	benzaldehyde	961	0.1	Shikimates and Phenylpropanoids
6	17.01	0.44	sabinene	979	0.1	Monoterpene
7	17.19	0.44	6-methyl-5-hepten-2-one	983	tr.	ketone
8	17.31	0.45	*β*-pinene	985	0.1	Monoterpene
9	17.59	0.45	myrcene	991	0.3	Monoterpene
10	18.52	0.48	*α*-phellandrene	1010	0.9	Monoterpene
11	19.12	0.49	*α*-terpinene	1022	0.2	Monoterpene
12	19.45	0.50	*p*-cymene	1029	0.3	Monoterpene
13	19.75	0.51	limonene	1035	4.3	Monoterpene
14	19.84	0.51	*β*-phellandrene	1036	0.6	Monoterpene
15	20.42	0.53	(*E*)-*β*-ocimene	1048	tr.	Monoterpene
16	20.6	0.53	dihydrotagetone	1052	0.1	Acyclic monoterpene
17	21.17	0.54	*γ*-terpinene	1063	0.4	Monoterpene
18	21.61	0.56	*Cis-*sabinene hydrate (thujanol or 4-thujanol)	1072	0.4	Monoterpene
19	22.48	0.58	2-nonanone	1089	9.7	ketone
20	22.7	0.58	terpinolene	1094	0.4	Monoterpene
21	22.87	0.59	2-nonanol	1097	0.8	fatty alcohols
22	22.97	0.59	linalool	1099	1.2	Monoterpene
23	23.1	0.59	nonanal	1102	0.2	saturated fatty aldehyde
24	23.8	0.61	phenylethyl alcohol	1116	0.1	Volatile alcohol
25	24.48	0.63	allo-ocimene	1130	0.1	Acyclic Monoterpene
26	25.19	0.65	(*E*)-tagetone	1145	0.2	unsaturated ketone
27	25.65	0.66	camphor	1154	4.4	Monoterpene
28	26.7	0.69	isoborneol	1176	0.1	Monoterpene
29	27.18	0.70	terpinen-4-ol	1185	0.2	Monoterpene
30	27.42	0.71	2-decanone	1190	0.8	Methyl ketone
31	27.56	0.71	dill ether	1193	0.7	Benzofurans
32	27.73	0.71	*α*-terpineol	1197	0.1	Monoterpene
33	28.02	0.72	cis-dihydrocarvone	1203	0.4	Monoterpene
34	28.42	0.73	trans-dihydrocarvone	1211	0.4	Monoterpene
35	28.92	0.74	shisofuran	1222	0.1	Monoterpene
36	29.41	0.76	2-nonyl acetate + cis-ocimenone	1232	2.3	Ester + Monoterpene
37	29.83	0.77	trans-ocimenone	1241	0.3	Monoterpene
38	30.23	0.78	carvone	1250	1.2	Monoterpene
39	30.78	0.79	piperitone	1261	0.2	Monoterpene
40	32.23	0.83	2-undecanone + bornyl acetate	1292	12.2	dialkyl ketone + acetate ester Monoterpene
41	32.52	0.84	2-undecanol	1299	0.4	Saturated fatty alcohol
42	32.91	0.85	(*Z*)-methyl cinnamate	1307	tr.	Phenylpropanoids
43	34.83	0.89	piperitenone	1350	2.0	Monoterpene
44	35.04	0.90	terpinyl acetate	1355	0.1	Monoterpene
45	35.46	0.91	6-dodecanone	1364	0.5	Ketone
46	35.84	0.92	piperitenone oxide	1373	0.2	Monoterpene
47	36.08	0.93	(*Z*)-ethyl cinnamate	1378	0.4	Phenylpropanoids
48	36.52	0.94	(*E*)-methyl cinnamate	1388	0.2	Phenylpropanoids
49	36.64	0.94	biphenyl (I.S.)	1391	3.2	Shikimates and Phenylpropanoids
50	37.13	0.96	methyl eugenol	1402	0.2	Aromatic ether (Phenylpropanoid)
51	37.27	0.96	*β*-elemene	1405	0.2	Sesquiterpenoids
52	38.04	0.98	cyperene	1424	0.1	Sesquiterpenoids
53	38.24	0.98	2-undecyl acetate	1428	0.4	Ester
54	38.85	1	(*E*)-*β*-caryophyllene	1443	24.1	Sesquiterpenoids
55	39.12	1.01	*β*-ylangene	1449	0.4	Sesquiterpenoids
56	39.29	1.01	geranyl acetone	1453	0.1	Monoterpene ketone
57	39.63	1.02	aromadendren	1461	0.5	Sesquiterpenoids
58	40	1.03	(*Z*)-ethyl cinnamate	1470	1.2	Phenylpropanoids
59	40.24	1.04	*α*-humulene	1476	0.5	Sesquiterpenoids
60	41.34	1.06	germacrene D	1502	1.3	Sesquiterpenoids
61	41.69	1.07	(*E*,*E*)-*α*-farnesene	1511	1.1	Acyclic Sesquiterpenoids
62	41.89	1.08	ledene	1516	0.6	Sesquiterpenoids
63	41.98	1.08	bicyclogermacrene	1518	0.4	Sesquiterpenoids
64	42.57	1.09	*γ*-cadinene	1533	0.7	Sesquiterpenoids
65	42.81	1.10	delta-cadinene	1539	1.3	Sesquiterpenoids
66	43.29	1.11	cadina-2,4-diene	1551	0.1	Sesquiterpenoids
67	43.49	1.12	*α*-cadinene	1556	0.2	Sesquiterpenoids
68	43.73	1.13	*α*-calacorene	1562	tr.	Sesquiterpenoids
69	43.98	1.13	(*E*)-nerolidol	1568	0.9	Sesquiterpene alcohol
70	44.99	1.16	cis-davanone	1593	1.8	Sesquiterpene
71	45.26	1.16	spathulenol	1600	1.0	Sesquiterpene
72	45.58	1.17	caryophyllene oxide	1608	0.3	Sesquiterpenoids
73	46.5	1.20	dill apiole	1632	0.3	Benzodioxoles
74	49.85	1.28	davanone-2-ol	1722	0.1	Phenylpropene
75	52.34	1.35	ethyl myristate	1791	0.1	Fatty acid ethyl ester
76	54.09	1.39	neophytadiene	1841	0.6	Acyclic diterpene
77	54.25	1.40	phytone (fitone)	1846	0.1	Acyclic Sesquiterpe
78	56.88	1.46	methyl palmitate	1922	0.2	Fatty acid methyl ester
79	59.2	1.52	ethyl palmitate	1990	0.2	Fatty acid ethyl ester
				SUM	**89.8%**	

**tr.**: Trace (<0.05%); **RRT**: Relative retention time and relative to (*E*)-*β*-caryophyllene (Rt = 38.85 min); **RI**: Refractive index; **NP**: Natural products.

**Table 2 molecules-27-07242-t002:** Antioxidant activity as EC_50_ (μg/mL) for DPPH and NO assay or FE (μg/mL) for FRAP assay of *T*. *patula* EO extracts (mean ± SD).

Treatments	DPPH (µg/mL)	Nitric Oxide (µg/mL)	FRAP FE (μg/mL)
**EO *T. patula***	29.85 ± 4.53	33.19 ± 3.8	30.22 ± 2.12
**Ascorbic Acid**	21.52 ± 2.02	34.63 ± 1.57	35.01 ± 2.59

DPPH: 2.2-Diphenyl-1-picrylhydrazyl; FRAP: Ferric reducing antioxidant power assay; EC_50_ of extracted EO against ascorbic acid.

**Table 3 molecules-27-07242-t003:** LD_50_ determination according to the Karbar method.

Group	Number of Animals	Dose (mg/kg)	Dose Differences (a)	Dead (*n*.)	Mean of Mortality (b)	Product (a × b)
1	6	10	-	0	-	-
2	6	20	10	0	0	0
3	6	50	30	1	0.5	15
4	6	100	50	2	1.5	75
5	6	200	100	3	3.5	350
						440

LD_50_: 200 − (440/6) = 126.667 mg/kg BW (approx.).

## Data Availability

Not applicable.

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
