# Peer review of "Chemical Composition of Tagetes patula Flowers Essential Oil and Hepato-Therapeutic Effect against Carbon Tetrachloride-Induced Toxicity (In-Vivo)"

_molecules, 2022, doi:10.3390/molecules27217242_

Round 1

Reviewer 1 Report

All the scientific names and in vitro should be written in italics.

In table 1 is indicated that E-β-caryophyllene is 24 % and the mauscript is indicated that it is 26.56 %, there is an inconsistency.

Figures 2, 3 and 4 are not required and should be removed.

Conclusions are results, so they need to be rewritten

Author Response

We are delighted to consider our work after revision and much thankful for the editor’s and all reviewers’ diligent effort in helping to improve our manuscript. The following document clarifies the inquiries raised during the reviewing cycle and provides our responses. The reviewers’ comments were considered, and a detailed clarification is addressed herein.

Reviewer 1

No.

Comments

Response

1.

All the scientific names and in vitro should be written in italics.

Done

2.

Table 1 is indicated that E-β-caryophyllene is 24 % and the manuscript is indicated that it is 26.56 %, there is an inconsistency.

For illustration: We calculated the % of each compound present at 89.9 % (i.e. we considered 89.9 as 100 % of our calculations). So, the result of 24.1 / 89.9 approx. is equal to 26 %.

To avoid conflict we standard only one style of writing through the manuscript.

Done

3.

Figures 2, 3, and 4 are not required and should be removed.

Done

Thanks for your valuable comment, to avoid conflict we removed figure 3. Also, we wish to fix figures 2 and 4 which are important to explain our study.

4.

Conclusions are results, so they need to be rewritten

Done

Reviewer 2 Report

The study presented by authors is very interesting, it describes the role of essential oil obtained from Tagetes patula flowers in hepatoprotection.

The work is part of the concerns of most researchers in the field of phytochemistry, to find new sources of compounds with hepatoprotective action.

The authors studied a species from the flora of the Kingdom of Saudi Arabia, a species recognized for its therapeutic virtues, studied and cited in the specialized literature.

The novelty of the study is the determination of the hepatoprotective role of Tagetes patulae aetheroleum.

The aim of this study was to characterize the volatile fraction obtained from the flowers of Tagetes patula which were collected from spontaneous flora of the Kingdom of Saudi Arabia. It is well known how pedoclimatic conditions influence the biosynthesis of biologically active compounds, including the chemical profile of the volatile oil.

The objective study is to determine the protection induced by the essential oil at the liver cell level, using experimental models on laboratory animals, in vivo tests.

To achieve all these goals, the authors describe the analytical and experimental methods used, interpret all the results, are discussed in detail and correlated with the specialized literature.

The introduction provides general information about the role and functions of a very vital organ that is always subject to various aggressions (chemical, metabolic, etc.), as well as theoretical aspects about the therapeutic importance of the flowers and essential oil of Tagetes patula.

All the working protocol is presented in the chronological order of the research, and the authors present all the working methods used. The authors use modern methods for the extraction and analysis of the essential oil, they also establish the extraction yield of the essential oil, knowing very well that the species Tagetes patula is not among the species very rich in essential oil.

To determine the hepatoprotective effects, the authors present the experimental methodology used, the experimental groups are correctly prepared, so that the direct effect of the essential oil can be followed compared to the group intoxicated with CCl4 and respectively the group where silymarin was used as a hepatoprotection. All the experimental protocols are presented, they respect all the imposed bioethical requirements.

The histopathological evaluation is very important, the authors also use photomicrographs as evidence.

A series of biomarkers responsible for oxidative stress induced at the hepatocyte level as well as serum markers of liver functions are determined. The authors perform analyses including for the purpose of determining the lipid profile.

The toxicological profile is also determined, with a very important role in research.

The antioxidant action of the essential oil is also determined, the authors using 3 methods of analysis (DPPH, NO, FRAP).

All the obtained results are interpreted statistically.

The results and discussions are punctual, for each type of analysis performed. Also in this chapter, the authors include tables, diagrams, spectra, graphs, very useful in the punctual evaluation of the conducted research.

The conclusions are relevant and succinctly present the results obtained.

The bibliography supports the ongoing research.

I agree with the publication in the form presented by the authors.

Author Response

We are delighted to consider our work after revision and much thankful for the editor’s and all reviewers’ diligent effort in helping to improve our manuscript.

We wish to thank him for his positive response.

Reviewer 3 Report

I want to congratulate the authors, this is an interesting paper, the study is relevant and the authors contribute to this field of research. Just a small observation I want to make: the authors need to check the list of references and correct it, according to the Instructions for authors.

Author Response

We are delighted to consider our work after revision and much thankful for the editor’s and all reviewers’ diligent effort in helping to improve our manuscript. The following document clarifies the inquiries raised during the reviewing cycle and provides our responses. The reviewers’ comments were considered, and a detailed clarification is addressed herein.

Reviewer 3

1.

I want to congratulate the authors, this is an interesting paper, the study is relevant and the authors contribute to this field of research. Just a small observation I want to make: the authors need to check the list of references and correct it, according to the Instructions for authors.

Done

Thanks a lot for your positive opinion and we checked the list of references according to the molecules journal